# Location-dependent maintenance of intrinsic susceptibility to mTORC1-driven tumorigenesis

Gabrielle V Rushing[1] , Asa A Brockman[1], Madelyn K Bollig[1] , Nalin Leelatian[1] , Bret C Mobley[2] , Jonathan M Irish[1,2] , Kevin C Ess[1,3] , Cary Fu[3] , Rebecca A Ihrie[1,4]

Neural stem/progenitor cells (NSPCs) of the ventricular–subventricular zone (V-SVZ) are candidate cells of origin for many brain tumors. However, whether NSPCs in different locations within the V-SVZ differ in susceptibility to tumorigenic mutations is unknown. Here, single-cell measurements of signal transduction intermediates in the mechanistic target of rapamycin complex 1 (mTORC1) pathway reveal that ventral NSPCs have higher levels of signaling than dorsal NSPCs. These features are linked with differences in mTORC1-driven disease severity: introduction of a pathognomonic *Tsc2* mutation only results in formation of tumor-like masses from the ventral V-SVZ. We propose a direct link between location-dependent intrinsic growth properties imbued by mTORC1 and predisposition to tumor development.

## Introduction

The ventricular–subventricular zone (V-SVZ) is the largest stem cell niche in the mammalian brain and is active during early postnatal development in both human and mouse (Guerrero-Cazares et al, 2011; Sanai et al, 2011). Neural stem/progenitor cells (NSPCs) within the V-SVZ have a positional identity—their dorsal–ventral and medial–lateral location within this spatially extensive stem cell niche predicts the type of neurons produced and correlates with the expression of region-specific transcription factors (Merkle et al, 2007, 2014; Young et al, 2007; Llorens-Bobadilla et al, 2015). However, whether dorsal versus ventral NSPCs have stereotypic signal transduction patterns or differential contributions to neurologic disease is unknown. We hypothesized that positionally linked features predispose cells to differing behaviors when disease-associated mutations occur.

The mechanistic target of rapamycin complex 1 (mTORC1) is a central regulator of cell size and growth. Within the V-SVZ, signaling via mTORC1 has been proposed to regulate self-renewal, proliferative divisions, differentiation, and brain ventricle morphogenesis (Paliouras et al, 2012; Foerster et al, 2017; Baser et al, 2019). In the developmental disorder tuberous sclerosis complex (TSC), patients carry mutations in either *TSC1* or *TSC2*, leading to an increase in mTORC1 signaling and tumor formation throughout the body (Crino et al, 2006). In the brain, approximately 80% of patients develop small periventricular tumors called subependymal nodules and 15% develop larger, potentially lethal tumors termed subependymal giant cell astrocytomas (SEGAs) (Krueger et al, 2013; Northrup et al, 2013). SEGAs frequently present near the foramen of Monro, a structure which includes the ventral area of the V-SVZ (Katz et al, 2012; Louis DN, 2016). Subependymal nodules and SEGAs are thought to originate from neural stem cells (Ess et al, 2005; Zhou et al, 2011; Feliciano et al, 2012), but the biological mechanisms underlying preferential SEGA localization are unknown. One possible etiology of SEGAs is a distinct ventral cell of origin. Here, patient tumors and mouse models were examined to investigate whether ventral subpopulations of stem cells exhibit a proliferation advantage and are uniquely capable of forming tumors when TSC2 is lost. Fluorescence flow cytometry was used to probe the intrinsic signaling capabilities of dorsal and ventral NSPCs with specific focus on mTORC1-dependent signaling events known to be disrupted in TSC.

## Results and Discussion

### SEGA samples from human TSC patients preferentially express the ventral transcription factor NKX2.1

To determine whether patient tumors express factors typical of specific V-SVZ subregions, both periventricular SEGA specimens and cortical hamartomas (tubers) from TSC patients were examined. Empty spiracles homeobox 1 (EMX1), which is expressed in developing cortex and in stem cells of the dorsal V-SVZ, was abundant in tubers (Fig 1A). By contrast, the ventral transcription factor NK2 homeobox 1 (NKX2.1) was not observed in these samples

[1]Department of Cell and Developmental Biology, Vanderbilt University School of Medicine, Nashville, TN, USA   [2]Department of Pathology, Immunology, and Microbiology, Vanderbilt University School of Medicine, Nashville, TN, USA   [3]Department of Pediatrics, Vanderbilt University Medical Center, Nashville, TN, USA   [4]Department of Neurological Surgery, Vanderbilt University Medical Center, Nashville, TN, USA

Correspondence: rebecca.ihrie@vanderbilt.edu

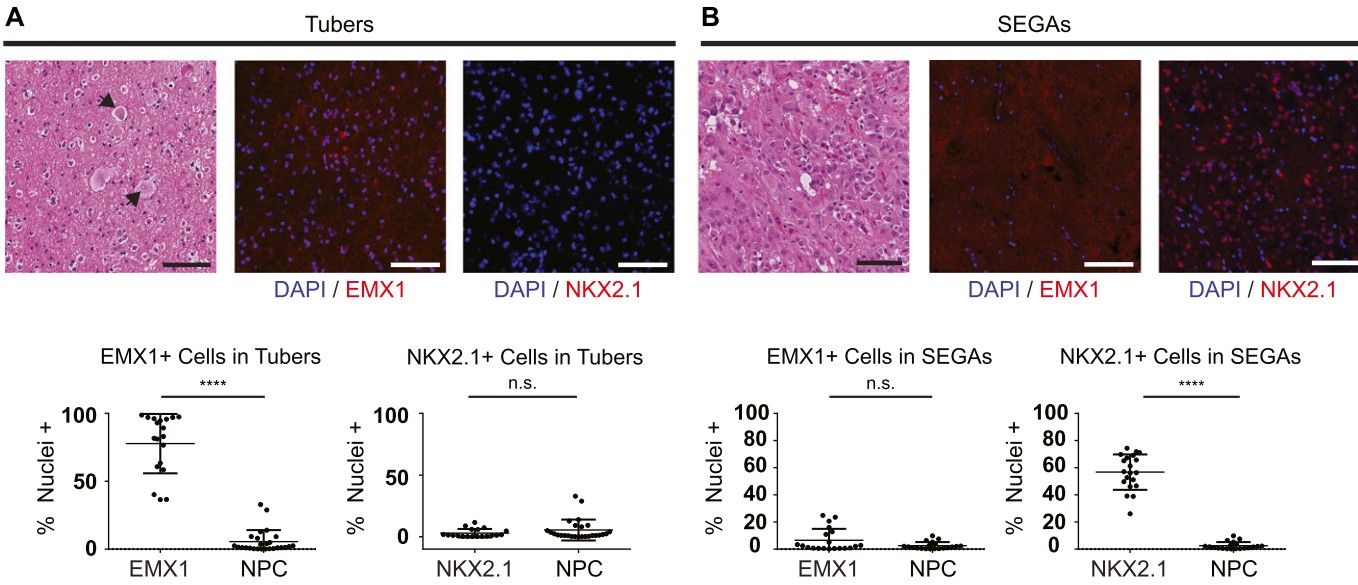

**Figure 1. Transcription factors defining dorsal or ventral identity are differentially expressed in dorsal and ventral hamartomas in TSC.**
**(A)** Representative fields from tilescans of human TSC cortical tubers stained with hematoxylin and eosin (left) or for DAPI (blue) and EMX1 (center, red) or NKX2.1 (right, red). Note the enlarged balloon cells (arrows) typical of tubers. Quantification of % positive nuclei for each factor is shown below. EMX1 is abundantly expressed (*P* < 0.0001 versus NPC), whereas NKX2.1 is not (*P* = 0.5809). N = 4 tubers, each dot = 1 region of interest (ROI), 4–5 ROIs/tuber. **(B)** Representative fields from tilescans of human SEGA tumors stained with hematoxylin and eosin (left) or for DAPI (blue) and EMX1 (center, red) or NKX2.1 (right, red). Quantification of positive nuclei is shown below, as in (A). In these ventral tumors, EMX1 is not widely expressed (*P* = 0.3373), but NKX2.1 is abundant (*P* < 0.0001). N = 4 SEGAs, each dot = 1 ROI, 5 ROIs/SEGA. Mann–Whitney tests were used. All scale bars = 100 *μ*m.

but was abundant in resected SEGA tissue (Fig 1B), consistent with prior case studies of transcript expression (Hewer & Vajtai, 2015; Hang et al, 2017). EMX1 was largely absent in SEGAs, suggesting that these two differentially localized malformations in TSC patients originate from distinct progenitor populations and that periventricular tumors are enriched for ventral markers.

### Ventral stem and progenitor cells have higher mTORC1 signaling than their dorsal counterparts

To analyze per-cell mTORC1 activity in V-SVZ subpopulations, dorsal and ventral NSPCs were dissected from neonatal mice and cultured as monolayers (Fig S1A). The cultures were first validated by measuring transcripts expressed in the dorsal (*Pax6*) or ventral V-SVZ (*Nkx2.1* and *Nkx6.2*) (Hack et al, 2005; Kohwi et al, 2005; Waclaw et al, 2006; Merkle et al, 2014; Delgado & Lim, 2015; Delgado et al, 2016). As expected, NSPCs were enriched for transcription factors present in their V-SVZ region of origin at both the transcript and protein level (Fig S1B and C).

Cultured NSPCs were used for flow cytometric measurement of phosphorylation events downstream of mTORC1 after gating for live, intact single cells (Fig S2A) (Hsu et al, 2011; Saxton & Sabatini, 2017). Known mTORC1 targets eukaryotic translation initiation factor 4E-binding protein 1 (p-4EBP1 T37/46) and ribosomal S6 protein (p-S6 S240/244) were phosphorylated at increased levels (e.g., a difference of 0.4 in the arcsinh-transformed median fluorescence intensity values, equivalent to an approximately twofold increase) in ventral cells (Fig 2A). Similarly increased levels of phosphorylated signal transducer and activator of transcription 3 (p-STAT3 S727), which is downstream of both the MAPK and mTORC1 pathways, were

also observed in ventral cells (Fig 2A). Dependence of these signaling pathways on mTORC1 was confirmed by treatment with rapamycin (Fig 2B). Consistent with the role of this pathway in regulating cell size and translation, ventral cells displayed small but significant differences in forward scatter by flow cytometry, indicating larger median size (Fig S2D). In addition, labeling with O-propargyl-puromycin (OPP) to detect newly translated proteins was elevated in ventral NSCs demonstrating increased protein synthesis (Fig 2C). Phosphorylation events not exclusively or specifically regulated by mTORC1, including p-S6 S235/236 (Fig 2A), p-PLCγ Y759, and p-ERK1/2 T202/Y204 (Fig S2C), did not differ significantly between dorsal and ventral cells. Similarly, total (unphosphorylated) levels of 4EBP1, STAT3, or S6 protein were not different between dorsal and ventral cells (Fig S2B). Levels of phosphorylated p38 mitogen-activated protein kinase (p-p38 MAPK T180/Y182) were higher in dorsal NSPCs, and p-Akt S473, upstream of mTORC1, exhibited a nonsignificant trend towards higher levels in dorsal NSPCs, likely because of feedback from mTORC2 (Fig S2C). These differences were retained across multiple passages (data not shown), consistent with previous findings demonstrating maintenance of regional identity and transcription factor expression through at least five passages (Delgado et al, 2016). Addition of media conditioned by the opposite cell type did not affect basal mTORC1 signaling, indicating that these differences are likely not due to differing autocrine stimulation (Fig S2E). Increased mTORC1 activity in ventral cells corresponded with slightly more population doublings per day when compared with their dorsal counterparts (Fig 4A). Critically, tuberin loss, via in vitro Cre transduction of cultures derived from *Tsc2*^fl/fl animals, resulted in a much greater difference between dorsal and ventral cells, with ventral cells

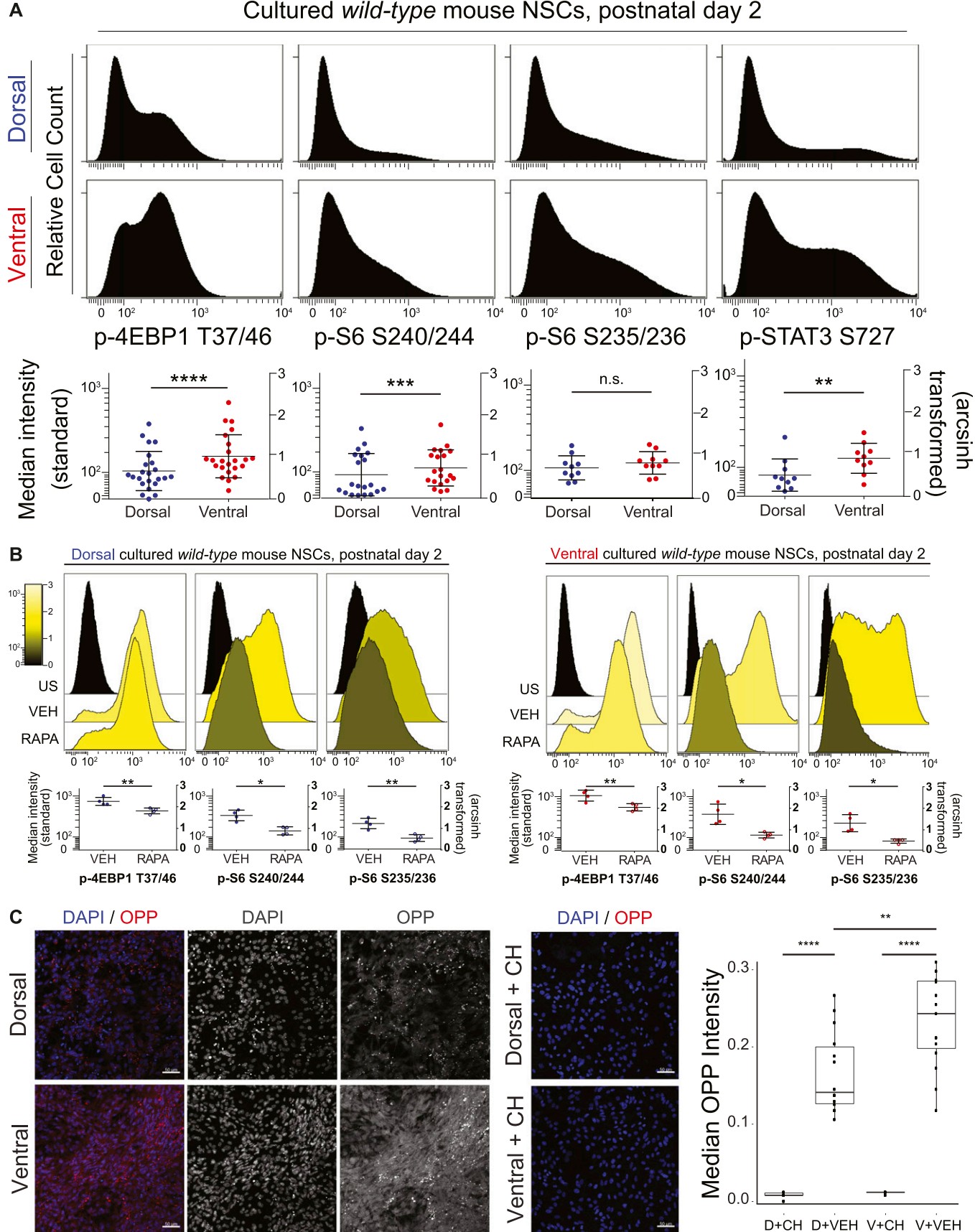

exhibiting approximately twice as many doublings per day (Fig 4B). These data indicate that ventral cells have a disproportionate proliferative advantage when mTORC1 signaling is disinhibited. Correspondingly, phosphorylation of 4EBP1 increased in both dorsal and ventral cultures upon tuberin removal, but dorsal–ventral differences remained (Fig 4C).

### Elevated ventral mTORC1 signaling is most evident in transit-amplifying cells

In addition to quantification in cultured progenitor cells, established protocols for acute dissociation and flow cytometric analysis of adult (P30) V-SVZ tissue were adapted to include fixation and measurement of intracellular antigens (Codega et al, 2014; Leelatian et al, 2017; Pastrana et al, 2009) (Fig 3A). Sequential biaxial gating was used to identify live intact single cells and specific V-SVZ cell types (Fig 3B). Cells collected from the ventral V-SVZ showed increased levels of p-4EBP1 T37/46 in the transit amplifying progenitor cell (TAP) population when compared with dorsal V-SVZ cells (Fig 3B and C), although differences in p-S6 S240/244 were not significant (Fig S3A). Surprisingly, activated neural stem cells (aNSCs) from the dorsal V-SVZ displayed slightly elevated levels of p-S6 S240/244 when compared with ventral aNSCs using this approach (Fig S3A), whereas no significant differences between dorsal and ventral were observed in other populations.

To complement these flow cytometric analyses, immunofluorescence was used to localize phosphoproteins within the intact V-SVZ of postnatal day-30 wild-type mice. Doublecortin (DCX)-positive neuroblasts were excluded as these migratory cells may have had either a dorsal or ventral origin (Lois & Alvarez-Buylla, 1994; Sawamoto et al, 2006). Robust staining of p-S6 S240/244 within the ventral V-SVZ was observed with significantly less abundant and intense staining in the dorsal niche (Fig S3B). Perinatal human brain V-SVZ exhibited a similar pattern of staining, with the ventral region displaying increased p-S6 S240/244 (Fig S3C). To identify TAP cells, measurement of phosphoprotein expression was then limited to Mash1-positive cells in postnatal day-30 wild-type mice. In these analyses, ventral V-SVZ cells had higher per-cell levels of both p-4EBP1 T37/46 and p-S6 S240/244 (Fig 3D and E).

mTORC1 signaling has been proposed as a key regulator of the activation of adult neural stem cells and expansion of the transit-amplifying progenitor pool (Paliouras et al, 2012). Prior reports have variably identified cells with detectable mTORC1 activity as transit-amplifying progenitors (C cells) and GFAP+ cells, including B1 stem cells (Paliouras et al, 2012; Hartman et al, 2013). Here, mTORC1 signaling was observed across the V-SVZ lineage, but differences in mTORC1 signaling between dorsal and ventral subpopulations were most evident within the TAP population. Consistent with these results, others have observed this progenitor population to have the highest level of protein synthesis compared with other lineage stages (Baser et al, 2019). High signaling in ventral cells was specific to the mTORC1 pathway, with p-4E-BP1 displaying the most robust difference in multiple approaches. This elevated phosphoprotein abundance was not observed across all proliferative signaling pathways, and p-p38 MAPK was elevated in dorsal progenitors, a feature not previously appreciated. These data reveal further potential modes of proliferative signaling outside of the mTORC1 pathway that may be dorsally enriched and highlight the selective enhancement of mTORC1 activity in ventral cells.

### Early tumor-like lesions develop exclusively after ventral-specific removal of *Tsc2* in the mouse

We hypothesized that the increased basal levels of mTORC1 signaling in ventral cells would lead to increased susceptibility to tumor formation versus dorsal counterparts. To directly test whether ventral NSPCs preferentially drive tumor development in a model of TSC, subregion-specific modulation of mTORC1 pathway activity was achieved using a conditional allele of *Tsc2* (Fu & Ess, 2013) in combination with the V-SVZ subregion-specific Cre alleles *Emx1^Cre* (Kessaris et al, 2006; Young et al, 2007) or *Nkx2.1^Cre* (Xu et al, 2008) to ablate tuberin expression in the dorsal or ventral V-SVZ, respectively. EMX1 is expressed in the developing telencephalic cortex (Gorski et al, 2002) and labels dorsal aspects of the adult V-SVZ comprising approximately 35% of the total niche (Gorski et al, 2002; Young et al, 2007). In contrast, NKX2.1 is expressed in the ventral embryonic forebrain (Sussel et al, 1999) and persists in the ventral-most tip of the lateral ventricle, contributing to approximately 5% of the total niche (Young et al, 2007; Lopez-Juarez et al, 2013; Merkle et al, 2014; Delgado & Lim, 2015). Both Cre drivers resulted in efficient loss of tuberin in targeted regions, with 79% or more of targeted V-SVZ cells exhibiting no detectable protein (Fig S4A–D), consistent with prior reports (Fu & Ess, 2013).

**Figure 2. Ventral mouse V-SVZ cells exhibit higher mTORC1 activity than dorsal cells and are responsive to rapamycin.**
**(A)** Representative histograms (from individual cultures) and graphs of median fluorescence intensity, relative to unstained samples, of phosphorylated proteins measured by flow cytometry. Dorsal: top and ventral: bottom. Each dot on graph represents an independent biological replicate (one culture from a P2 mouse). Left scale: $\log_{10}$ and right scale: arcsinh transformed values. Events indicating mTORC1 activity are significantly elevated in ventral cells (p-4EBP1 T37/46, $P < 0.0001$; p-S6 S240/244, $P = 0.0005$; p-STAT3 S727, $P = 0.0026$), whereas other events do not differ (p-S6 S235/236, $P = 0.1617$). A difference of 0.4 (as in the case of p-4EBP1) on the arcsinh scale represents an approximately twofold difference in total phosphorylated epitope levels per cell. Paired $t$ tests were used. **(B)** Representative histograms (top) and graphs (bottom) of unstained (US), vehicle-treated (VEH), and rapamycin-treated (RAPA) dorsal (left) and ventral (right) cultures for the indicated phosphoproteins. The median fluorescence intensity for each phosphoprotein is graphed using a standard $\log_{10}$ scale (left) and the arcsinh-transformed scale (right), with both vehicle and rapamycin-treated samples shown relative to unstained cells. Both dorsal and ventral cultured NSPCs respond to rapamycin treatment (30 nM, 24 h) showing decreased signal compared with vehicle (30 nM DMSO, 24 h). Paired $t$ tests were conducted comparing VEH with RAPA: dorsal p-4EBP1 T37/46 ($P = 0.0083$), p-S6 S240/244 ($P = 0.0309$), p-S6 S235/236 (0.0092); ventral p-4EBP1 T37/46 (0.0018), p-S S240/244 (0.0157), p-S6 S235/236 (0.0188). N = 4, each N represents cells cultured from an individual P2 mouse. **(C)** Representative images of dorsal and ventral cultured NSPCs stained for nuclei (DAPI, blue) and OPP (red), which labels newly translated proteins; left (merged image), right (individual grayscale images). Middle: representative images of dorsal and ventral cultured NSPCs pretreated for 30 min with 100 µg/ml cycloheximide (CH), an inhibitor of protein synthesis. Right: box and whisker plot showing quantification of median OPP pixel intensity for each condition (arbitrary units). Scale bars, 50 µm. N = 3 for vehicle-treated, N = 1 for CH-treated. Each N = cells from an individual P2 mouse. Repeated measures ANOVA was conducted in GraphPad Prism ($P < 0.0001$) followed by Sidak's multiple comparisons test. For all graphs, bars represent mean ± SD.

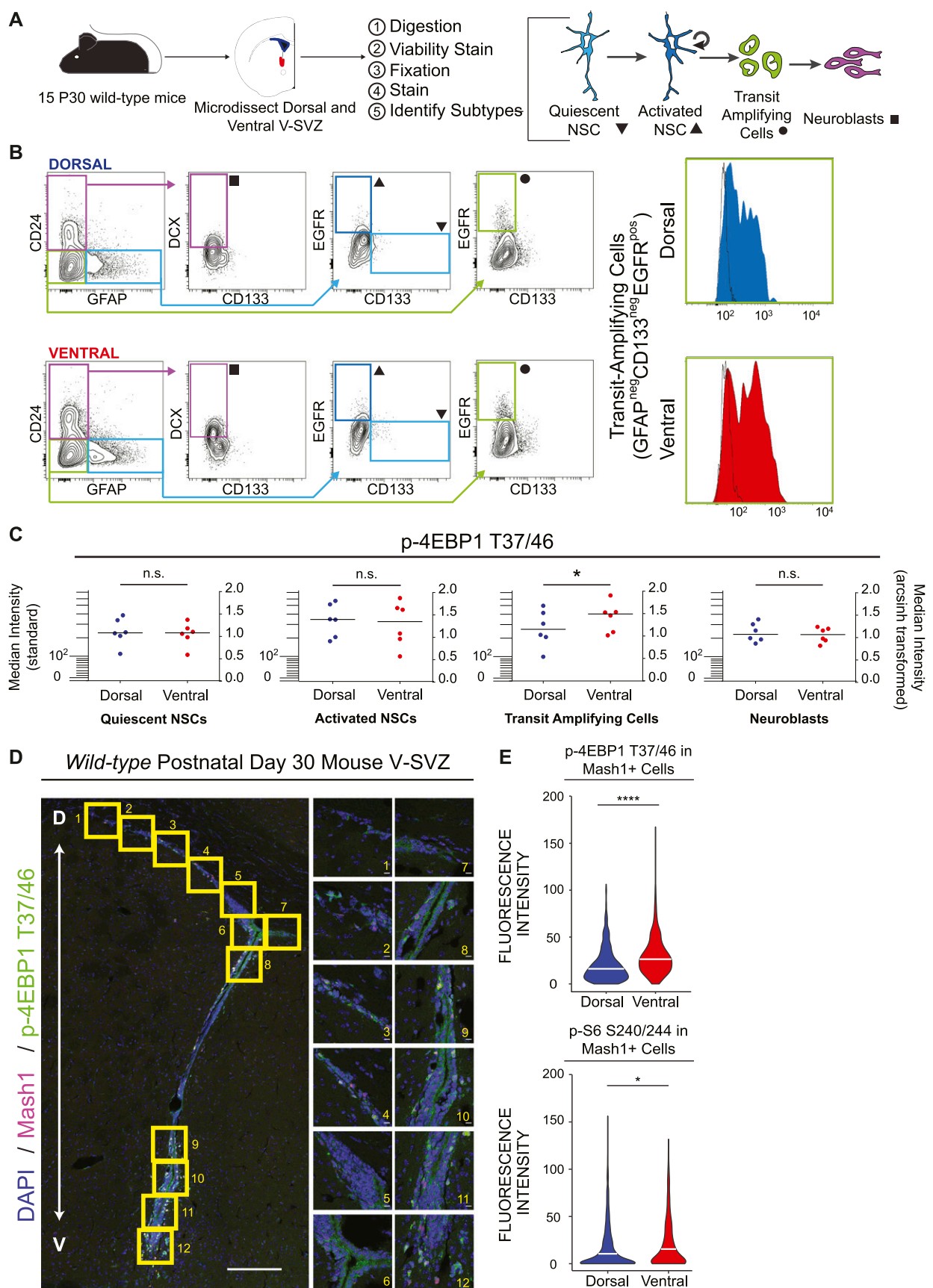

At P7, epilepsy and perinatal lethality were observed in *Emx1^Cre^; Tsc2^Fl/Fl^* mice (Fu & Ess, 2013) (data not shown), but tumor formation was not reproducibly evident (Fig 4E). By contrast, at the same time point, large aberrant GFAP-positive cellular collections were observed in the ventral V-SVZ in 63.6% (7/11) of *Nkx2.1^Cre^; Tsc2^Fl/Fl^* animals, with 27.3% (3/11) exhibiting larger tumor-like protrusions in the third ventricle (Fig 4F and G). These phenotypes were not found in controls (Fig 4D) or animals with dorsal loss of tuberin protein, with only one *Emx1^Cre^; Tsc2^Fl/Fl^* animal presenting with a small (<10 nuclei) cell cluster near the ventral V-SVZ (Fig 4H). Clusters of abnormal cells expressed variable high GFAP and p-S6 (Fig 4F and G) and exhibited densely packed, hyperchromatic nuclei (Fig S4E and F), similar to prior reported models and human SEGAs.

Until now, it has not been shown that differences within the same neural stem cell niche affect susceptibility to tumor development. Central questions within the example case of TSC have been whether SEGAs originate from smaller tumors and whether their larger size and stereotypic ventral presentation is due exclusively to microenvironmental factors—a model postulated in prior animal studies (Zhou et al, 2011). Previous TSC models removed the tuberin binding partner hamartin (encoded by *Tsc1*) throughout the embryonic or postnatal niche and observed structural abnormalities in the lateral ventricle (Zhou et al, 2011; Magri et al, 2013), but no studies have assessed the specific contributions of V-SVZ subpopulations to the development of these features. An additional mouse model co-deleted *Tsc1* and *Pten* in postnatal NSCs and found tumors that closely recapitulated human tumors via hematoxylin and eosin staining (Zordan et al, 2018); however, to date, no TSC patients have been reported to have a mutation in *Pten* (Martin et al, 2017). Although we cannot fully exclude the possibility that dorsal V-SVZ progenitors may also develop into nodules given additional time, the selective, robust generation of tumor-like growths from a relatively small region of the niche indicates a marked sensitivity to the effects of *Tsc2* mutation within ventral V-SVZ cells. The data here argue that periventricular tumors in TSC are largely due to the unique susceptibility of ventral NSPCs to mTORC1 dysregulation: only *Nkx2.1^Cre^; Tsc2^Fl/Fl^* animals reliably developed tumor-like cellular protrusions in the V-SVZ and third ventricle at the time points examined, and tuberin loss in cultured ventral cells increased their rate of doubling beyond that seen in matched dorsal cells. Coupled with the specific absence of EMX1 and frequent expression of NKX2.1 in human SEGAs, these findings indicate that ventral neural progenitors defined by NKX2.1 expression and elevated mTORC1 signaling are likely the cells of origin for large tumors in TSC.

Heterotopic transplantation experiments and the study of cultured progenitor cells have demonstrated that the transcriptional identity of V-SVZ cells is at least partially cell intrinsic (Merkle et al, 2007; Delgado et al, 2016). Transcription factors appear to be crucial in initiating or maintaining this feature, as forced expression of ventral transcription factors such as Gli1 can alter subsequent neuronal identity (Ihrie et al, 2011). Similarly, embryonic loss of Nkx2.1 results in respecification of cells from a ventral fate to a dorsal fate (Sussel et al, 1999). The differences in mTORC1 signaling identified here show that beyond transcriptomic heterogeneity, an additional level of functional heterogeneity is coupled to position within the V-SVZ. These findings raise the questions of whether identity-associated transcription factors are determinants of basal signaling levels and how early in neural development these signaling states might emerge. If signaling controls identity, it may be possible to productively modulate signaling to alter cell identity. Alternatively, if identity controls signaling, future therapeutic approaches could exploit this specificity to target particular pools of NSPCs in disease states without compromising the entire niche. Collectively, these findings suggest that brain tumors derived from different NSPC origins are distinct diseases requiring differential treatment regimens—a concept with application to other solid tumors, including malignant brain tumors and tumors in epithelia organized along dorsoventral or rostrocaudal coordinates.

# Materials and Methods

### Contact for reagent and resource sharing

Further information and requests for resources, protocols, and reagents should be directed to and will be fulfilled by the Lead Contact, Rebecca Ihrie (rebecca.ihrie@vanderbilt.edu).

### Animals

All animal procedures were carried out in accordance with institutional (Institutional Animal Care and Use Committee) and National Institute of Health guidelines. Mouse lines used were of a mixed background (primarily C57 BL/6J with 129S6 and CD1). Critically, for each mouse experiment, the controls were littermates and thus on an equivalent genetic background. If sufficient numbers of wild-type mice were not available in the colony for an experiment, wild-type C57 BL/6J mice were ordered from Charles River Laboratories. For cultures, P2 C57 BL/6J pups were used. For

---

**Figure 3. The TAP population exhibits the highest mTORC1 signaling differences between dorsal and ventral regions.**
**(A)** Cartoon schematic outlining prospective isolation from dissection to cell identification. **(B, C)** Biaxial gating strategy to obtain V-SVZ cell types. Symbols at terminal gates correspond to cell types in (C). To the right are representative histograms showing intensity of p-4EBP1 T37/46 signal in freshly isolated dorsal (blue) and ventral (red) TAPs. Black histogram outline shows the fluorescence minus one control for each sample. **(C)** Graphs show median fluorescence intensity data for p-4EBP1 T37/46 from 7 sets of 15 pooled mice each (105 total mice); left: standard scale and right: arcsinh-transformed. p-4EBP1 T37/46 is elevated in ventral TAPs relative to dorsal ($P = 0.0175$, paired $t$ test, bar = median) but is not different in the other cell types shown (qNSCs [$P = 0.1931$], aNSCs [$P = 0.5853$], and neuroblasts [$P = 0.1374$]). **(D)** Representative tilescan confocal images of V-SVZ stained for DAPI (blue), Mash1 (red), and p-4EBP1 T37/46 (green), with boxed areas highlighted to the right. Scale bars: 100 $\mu$m (tilescan), 10 $\mu$m (63× images). **(E)** Quantification of per-cell intensity for p-4EBP1 T37/46 (top: 1,026 dorsal and 590 ventral cells total) and p-S6 S240/244 (bottom: 1,306 dorsal and 805 ventral cells) in confocal images. Both phosphorylation events are elevated in ventral Mash1+ cells relative to dorsal ($P < 0.0001$ [p-4EBP1] and $P = 0.0161$ [p-S6 S240/244], Wilcoxon signed rank tests). Each experiment: n = 4 mice, 3 sections/mouse. For all graphs, bars represent mean ± SD.

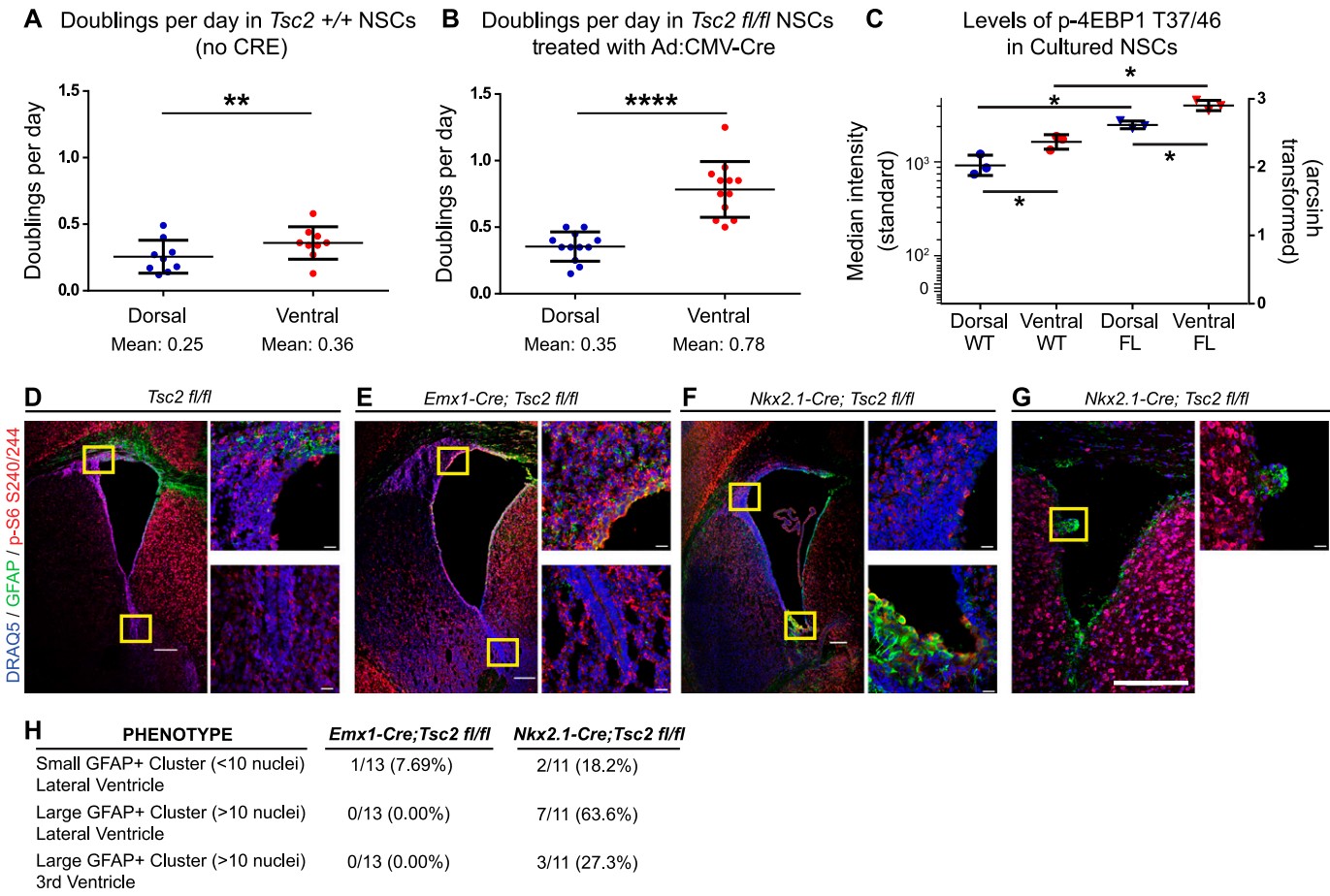

**Figure 4. Ventral-specific removal of *Tsc2* leads to more rapid doubling time, elevated mTORC1 activity, and abnormal growth in the V-SVZ.**
**(A)** Graph illustrating population doublings per day in *wild-type* dorsal and ventral cultures. Ventral cells exhibit slightly increased doubling times as compared with dorsal, consistent with previous findings (Delgado et al, 2016). N = 3 mice counted three times. Paired *t* test, P = 0.0018. **(B)** Graph illustrating doublings per day in *Tsc2*$^{fl/fl}$ dorsal and ventral cultures treated with Ad:CMV-Cre. When tuberin is lost, ventral cells double twice as fast as their dorsal counterparts do. N = 4 mice, counted four times. Paired *t* test, P = 0.0001. **(C)** Graph showing the median fluorescence intensity of p-4EBP1 T37/46 in *wild-type* and *Tsc2*$^{fl/fl}$ dorsal and ventral cultures (treated with Ad:CMV-Cre). High signaling is observed as compared with dorsal in *wild-type* ventral cultures. Upon *Tsc2* removal, both dorsal and ventral cultures exhibit higher p-4EBP1 T37/46 than *wild-type* but dorsal–ventral differences are maintained. Repeated measures ANOVA with Holm–Sidak's multiple comparisons test, P = 0.0005. N = 3 mice per condition. **(D–G)** 10-μm P7 brain sections stained for DRAQ5 (nuclei, blue), GFAP (green), and p-S6 S240/244 (red). **(D)** Representative confocal images of V-SVZ subregions are shown for *Tsc2*$^{fl/fl}$ control mouse. **(E)** *Emx1-Cre; Tsc2*$^{fl/fl}$ animals do not exhibit tumor growth. **(F, G)** Representative confocal images for *Nkx2.1-Cre; Tsc2*$^{fl/fl}$ reveal GFAP+ cellular protrusions in the ventral V-SVZ (F). **(G)** A larger tumor-like structure is apparent in the third ventricle (G). **(H)** Phenotypic summary of all mouse models. All images: scale bar = 200 μm and insets = 20 μm. For all graphs, bars represent mean ± SD.

experiments using wild-type adult animals, P30 male C57 BL/6J animals were used. For subregion-specific cre lines (*Emx1*$^{Cre}$; *Tsc2*$^{fl/fl}$ and *Nkx2.1*$^{Cre}$; *Tsc2*$^{fl/fl}$), P7 animals were analyzed because of perinatal lethality observed in the *Emx1*$^{Cre}$; *Tsc2*$^{fl/fl}$ animals (Fu & Ess, 2013). *Tsc2*$^{fl/fl}$ animals were a kind gift from Kevin Ess and Cary Fu (Vanderbilt University). *Emx1*$^{Cre}$ and *Nkx2.1*$^{Cre}$ animals were also kind gifts from Kevin Ess and Cary Fu, with original orders made from The Jackson Laboratory (https://www.jax.org/).

## Mouse genotyping

*Tsc2*: TGGCAGGACAGAGGGTCATCATGG (TSC2-Forward), TTCAGAGTC-ACCTGGCAGGCTCG (TSC2-Reverse);
 *Cre*: TAAAGATATCTCACGTACTGACGGTG (nlsCre-Forward), TCTCT-GACCAGAGTCATCCTTAGC (nlsCre-Reverse).

## Human tissue samples

De-identified GW33+0 brain tissue sections were provided by the University of California Pediatric Neuropathology Consortium. All human tuber and SEGA tissues were obtained with consent under Vanderbilt University Institutional Review Board #180238 and #130550. Authors attest that experiments conformed to the principles set out in the World Medical Association Declaration of Helsinki and the Department of Health and Human Services Belmont Report. Information on age at resection and clinical history is included in the table below.

## Primary cell cultures

For experiments using cultured V-SVZ cells, the cells were prepared from postnatal day-2 wild-type pups as previously described

## Resected tissue information.

| Case no. | Age at resection | Gender | Clinical history |
|---|---|---|---|
| 1 | 11 | Male | TSC, SEGA tumor, foramen of Monro |
| 2 | 12 | Male | TSC, SEGA tumor, left foramen of Monro |
| 3 | 17 | Male | TSC, SEGA tumor, right foramen of Monro |
| 4 | 3 | Female | TSC, SEGA tumor, left foramen of Monro |
| 5 | 2 | Female | TSC, cortical tuber tissue |
| 6 | 2 | Female | TSC, cortical tuber tissue |
| 7 | 2 | Male | TSC, cortical tuber tissue |
| 8 | GW33+0 | Not provided | Perinatal human brain tissue, COD not brain-related |

(Merkle et al, 2007; Delgado et al, 2016). In brief, dorsal and ventral V-SVZ tissue was minced and incubated with 0.25% Trypsin–EDTA for 20 min at 37°C, 5% $CO_2$ followed by mechanical dissociation and plating in N5 media. Cells were fed every other day and passaged upon confluence. Dorsal and ventral cells were passaged on the same days and plated at equivalent numbers. For experiments using $Tsc2$ $^{fl/fl}$ cells, Ad:CMV-Cre (Vector Biolabs) was added at passage 1 at a concentration of $1 \times 10^7$ pfu/ml. dTomato expression was confirmed 72 h after virus addition using epifluorescence microscopy and subsequently confirmed via flow cytometry.

### Flow cytometry—cultured cells

All files associated with this manuscript, including cultured cells and dissociated tissue, are available on FlowRepository under the ID FR-FCM-Z2ZZ (Spidlen et al, 2012). Cultured cells were collected for flow cytometry experiments at passage 2 using dissociation protocols as previously described (Irish et al, 2010). In brief, the cells were treated with Accutase for 15 min at 37°C, 5% $CO_2$ on a nutator followed by mechanical dissociation (trituration) into a single-cell suspension. The cells were then pelleted by centrifugation (5 min, 0.3 RCF) and resuspended in equivalent volumes of their original media in round bottom tubes (352052; Corning). The cells in tubes then rested at 37°C, 5% $CO_2$ for 2 h, with Alexa Fluor 700-SE dye (A-20010; Life Technologies) added for the last 15 min of incubation to label non-intact cells. Subsequently, the cells were fixed with a final concentration of 1.6% paraformaldehyde for 20 min at room temperature, washed with 1× PBS, and spun for 5 min at 800 $g$ at room temperature. The tubes were decanted and the cells were resuspended in the void volume by vigorous vortexing. Once resuspended, 100% cold methanol was added. The cells were incubated at −20°C for 20 min and either used immediately for flow cytometry experiments or stored at −80°C until staining. All antibody clones and dilutions are listed in Table S1.

The experiments were run on either BD LSR II or Fortessa 5-laser instruments. Single fluorophore–labeled beads and unstained cell lines were used as compensation and sizing controls. Signaling was quantified as the fold change in per-cell phosphoprotein median

fluorescence intensity of stained samples compared with unstained samples from the same experiment. In the case of cells with the dTomato RFP reporter (Ai14), stained samples were compared with reporter-only samples. The inverse hyperbolic sine (arcsinh) with a cofactor was used to compare samples as previously described (Irish et al, 2010). The arcsinh median of intensity value × with cofactor c was calculated as arcsinhc(x) = ln(x/c + √((x/c)² + 1)). The cofactor (c) is a fluorophore-specific correction for signal variance. All analyses were completed using Cytobank (cytobank.org) (Kotecha et al, 2010). Paired $t$ tests were used to compare samples.

### Flow cytometry—acute adult V-SVZ dissociation

In the case of acute adult V-SVZ dissociation, published dissociation protocols were adapted for mouse tissue (Leelatian et al, 2017). Periventricular tissues were dissected from 15 C57 BL/6J postnatal day-30 mice and pooled. Tissue pieces were minced with microknives, collected in 15-ml conical tubes (0553859B; Corning) using cold PIPES buffer, and centrifuged at 100 $g$ for 5 min at room temperature. Post spinning, the cells were dissociated in a total volume of 5 ml: 4.9 ml of DMEM-F12 + Glutamax (10565018; Thermo Fisher Scientific), 50 $\mu$l of 100× Collagenase II (6885; Sigma-Aldrich), and 50 $\mu$l of 100× DNase I (DN25; Sigma-Aldrich) by mixing followed by incubation at 37°C, 5% $CO_2$ on a nutator for 30 min. These conditions were chosen as they were optimal for cell viability during tissue dissociation (Leelatian et al, 2017). The resulting mixture was then strained through a 70-$\mu$m cell strainer followed by a 40-$\mu$m cell strainer using warm DMEM-F12 + GlutaMax media. The strained mixture was then centrifuged at 100 $g$ for 10 min at room temperature, and the supernatant was discarded. Pellet was resuspended in 5 ml warm DMEM-F12 + GlutaMax and centrifuged at 100 $g$, 10 min at room temperature, and the supernatant was discarded. The cells were treated with ACK lysis buffer for 1 min at room temperature followed by the addition of an equivalent volume of warm DMEM-F12 + GlutaMax media. This mixture was then centrifuged at 100 $g$, 10 min at room temperature, and the supernatant was discarded. Cell pellets were resuspended in warm DMEM-F12 + GlutaMax with Alexa Fluor 700-SE dye for 10 min to label non-intact cells. Subsequently, the cells were fixed with a final concentration of 1.6% paraformaldehyde for 20 min at room temperature, washed with 1× PBS, and spun for 5 min at 800 $g$ at room temperature. After washing, surface antibody staining was conducted for EGFR-biotin, CD133, and CD24 for 30 min to 1 h at room temperature in Brilliant buffer; cells were washed with 1× PBS-BSA; and then subjected to secondary staining with streptavidin-BV786 for 30 min to 1 h at room temperature in Brilliant buffer. The cells were washed with 1× PBS-BSA, spun for 5 min at 800 $g$ at room temperature, and resuspended in the void volume by vortexing after decanting. 1 ml of ice-cold 70% ethanol was added to cell pellets and the tubes were stored at −20°C for 20 min to permeabilize cells. The cells were then washed with 1× PBS-BSA and spun for 5 min at 800 $g$ at room temperature. Intracellular staining was conducted for DCX, GFAP, and either p-S6 S240/244 or p-4EBP1 T37/46 for 30 min to 1 h at room temperature. The experiments were run on a BD LSR II 5-laser instrument. Single fluorophore–labeled beads were used as compensation controls and fluorescence-minus-one tubes were created

for the phosphoprotein channel to assess background signal. Gating for V-SVZ population was completed using cell size controls for comparison (MV411 cells for smaller size and U87 cells for large size). All antibody clones and dilutions are listed in Table S1. All analyses were completed using Cytobank software (cytobank.org).

## V-SVZ culture-conditioned media experiments

Passage 2 V-SVZ NSPCs from P2 mice were plated at an equal number across all samples. Approximately 24 h after plating, a complete medium change was performed. 24 h later, the medium was swapped between wells of one set of dorsal and ventral cells from each sample using a 0.22-$\mu$m filter to prevent any cross-well cellular contamination. Control cells had media removed, filtered, and replaced in the same well. 24 h after media exchange, the cells were collected for flow cytometry. A paired $t$ test was performed to determine statistical significance.

## Immunostaining—mouse tissue

50–60-$\mu$m sections were cut using a Leica SM 2010R sliding microtome with a Physitemp freezing stage (P2–P7 mice at 60 $\mu$m, P30 and older at 50 $\mu$m) and stored at –20°C in 24-well dishes filled with antifreeze solution (Lu & Haber, 1992). Frozen P7 brains from *Tsc2* ablation studies were sectioned at 8-$\mu$m intervals by the Translational Pathology Shared Resource. Floating sections were incubated in blocking solution (PBS/1% normal donkey serum/1% BSA/0.1% Triton X-100) for 30 min at room temperature and then incubated with primary antibodies overnight at 4°C (see Table S1 for antibodies and dilutions). The sections were washed for 5 min three times with 1× PBS the following day and then incubated with secondary antibodies and DAPI (1:10,000) for 2–3 h at room temperature. The sections were washed for 5 min three times with 1× PBS and mounted on Coler Frost plus microscope slides (12-550-16; Thermo Fisher Scientific). Once dry, the slides were rinsed with ddH20 followed by the addition of mounting media (Mowiol) and a coverslip. Frozen sections on slides were treated the same; however, they were stored in a humidifier chamber during staining.

## Immunostaining—human tissue

Tissue sections embedded in paraffin were dry-baked vertically for 30 min at 55°C in Coplin jars and then cooled to room temperature. De-paraffinizing, rehydrating, and unmasking were completed using Trilogy buffer (920P-09; Cell Marque) and a 16-min antigen retrieval in a pressure cooker. The slides were washed with 1× PBS and incubated in blocking solution (PBS/1% normal donkey serum/ 1% BSA/0.1% Triton X-100) for 30 min at room temperature and then incubated with primary antibodies overnight at 4°C in a humidifier chamber (See Table S1 for antibodies and dilutions). No primary controls (NPCs) did not receive primary antibody on the transcription factor channel. The sections were washed for 5 min three times with 1× PBS the following day and then incubated with secondary antibodies and DAPI (1:10,000) for 1–2 h at room temperature. The sections were washed for 5 min three times with 1×

PBS and mounted on SuperFrost Plus Microscope Slides (12-550-16; Thermo Fisher Scientific). Once dry, the slides were rinsed with ddH2O followed by the addition of mounting media (Mowiol) and a coverslip.

## Image quantification—transcription factors in cultures

Dorsal and ventral V-SVZ subregions were dissected from six postnatal day-2 CD1 mice and cultured for two passages before plating on eight-well chamber slides. Dorsal and ventral cultures were grown to confluence before fixation and staining. The cells were stained with DAPI, PAX6 (BioLegend), and NKX2.1 (Santa Cruz) (see Table S1 for dilutions). Three fields per well were imaged using a Zeiss LSM710 confocal microscope. Transcription factor expression was quantified using FIJI (ImageJ), first by segmenting total nuclei per field, followed by PAX6 and NKX2.1-positive nuclei identification per field. The transcription factor percent positive was averaged across the three fields for each biological replicate. Finally, a paired $t$ test was used to compare transcription factor expression in dorsal versus ventral cultures across six biological replicates.

## Image quantification—mouse V-SVZ

20× confocal tilescan images and 63× confocal V-SVZ subregion images were obtained using the Zeiss LSM880 microscope. The freehand tool in ImageJ was used to identify positive or negative cells of interest on the lineage-identifying channel (DCX or Mash1), and the "Select → Restore Selection" tool was used to overlay the freehand cell outline onto the phosphoprotein channel being analyzed. Mean gray values (MGV) for the phosphoprotein channel of interest were recorded, and background fluorescence on the phosphoprotein channel was subtracted from each MGV value. MGV values for both dorsal and ventral were analyzed using GraphPad Prism software; Wilcoxon signed rank tests were used to compare dorsal and ventral values. Violin plots were generated using R-project software.

## Image quantification—human SEGAs and tubers

Whole slide imaging was performed at the Digital Histology Shared Resource at the Vanderbilt University Medical Center (www.mc.vanderbilt.edu/dhsr). For each SEGA and tuber sample, H+E, NPC, EMX1-stained, and NKX2.1-stained images were aligned to allow quantification of the same region across 5-$\mu$m serial sections. Within regions identified as idiopathic without high blood cell infiltration by a pathologist (BC Mobley), CellProfiler was used to segment DAPI+ nuclei across 4–5 1,300 × 1,300-$\mu$m fields per sample and to measure integrated intensity of NPC, EMX1, and NKX2.1 as well as the nuclear area for each identified nucleus. All data were imported into Cytobank (Cytobank.org) and EMX1+ and NKX2.1+ nuclei were gated by using the NPC nuclei as a negative control. The % positive signal for EMX1 and NKX2.1 was determined by gating on biaxial plots of integrated intensity versus area to account for any differences in the size of nuclei. Wilcoxon signed rank tests were used to analyze the results.

### Image quantification—OPP

For OP-puromycin labeling, the Thermo Fisher Scientific Click-iT Plus Protein Synthesis Assay with the Ax647 fluorophore was used according to the manufacturer's instructions. All images were acquired with identical settings on a Zeiss LSM880 confocal microscope. For analysis, cell nuclei were segmented in CellProfiler using DAPI staining, and median OPP pixel intensity was calculated for each segmented nucleus. Median nuclear OPP intensity per field was evaluated because of high confluence in cultures preventing accurate assignment of cytoplasmic regions to nuclei. In addition, OPP staining overlapped with nuclei in the vast majority of cases. The box and whisker plot was drawn in R as previously described (https://ggplot2.tidyverse.org/reference/geom_boxplot.html).

### Real-time quantitative reverse transcription PCR (qRT-PCR)

Total RNA from passage 3 dorsal and ventral cultures from 5 *wild-type* littermates were isolated using standard protocols (Ihrie et al, 2011). Transcript expression was measured in triplicate with Taqman assays (Applied Biosystems) for *Ubc* (Mm01201237_m1), *Nkx2.1* (Mm00447558_m1), *Nkx6.2* (Mm00807812_g1), and *Pax6* (Mm00443081_m1), using a CFX96 Real-Time System on a C1000 Thermal Cycler (Bio-Rad). Relative expression of transcripts was calculated in Excel using the delta–delta CT method with *Ubc* as an endogenous control. A paired *t* test in GraphPad Prism was used to determine statistical significance.

### Measurement of V-SVZ culture doubling times

After V-SVZ NSPCs were dissected and cultured from the dorsal and ventral regions of C57BL/6 (C57) wild-type mice, they were passaged every 4 d with equal numbers of live cells replated each time. At each passage, live cells were carefully counted using a hemacytometer and trypan blue staining. Five fields of view were counted per sample at each time point. A paired *t* test in GraphPad Prism was used to determine statistical significance in the doublings per day between the ventral and dorsal cells.

### Quantification and statistical analysis

All tests are specified in figures and figure legends. GraphPad Prism was used for all analyses unless otherwise specified. For sample size estimation in region-specific removal of *Tsc2*, the Power and Sample Size (PS) software (Vanderbilt Biostatistics resource) was used with the assumption of differences in tumor size or latency of 30% or more, an effect size smaller than size differences in published TSC models (Zhou et al, 2011; Feliciano et al, 2012).

## Supplementary Information

## Acknowledgements

The authors thank the Ihrie and Irish labs and Christopher VE Wright for helpful discussions on data and figure organization, Brittany Parker (Fu lab), Amanda Jurewicz, and Ethan Chervonski (Ihrie lab) for assistance with the collection of mouse samples, and David K Flaherty (VUMC Flow Cytometry Shared Resource) for helpful discussions. Confocal microscopy experiments were performed in part through the use of the Vanderbilt University Cell Imaging Shared Resource (supported by NIH grants S10 1S10OD021630-01, CA68485, DK20593, DK58404, DK59637, and EY08126). Whole slide imaging of TSC SEGA and tuber tissue was performed at the Digital Histology Shared Resource at Vanderbilt University Medical Center (www.mc.vanderbilt.edu/dhsr). Flow Cytometry experiments were performed in the Vanderbilt University Medical Center Flow Cytometry Shared Resource (supported by the Vanderbilt Ingram Cancer Center (P30 CA68485) and the Vanderbilt Digestive Disease Research Center (DK058404)). The Vanderbilt Translational Pathology Shared Resource is supported by NCI/NIH Cancer Center Support Grant 2P30 CA068485-14 and the Vanderbilt Mouse Metabolic Phenotyping Center Grant 5U24DK059637-13. This work was supported by NIH 2T32CA009592 (GV Rushing), F31 NS096908 (GV Rushing), NIH/NINDS NS096238 (RA Ihrie), NCI R00 CA143231 (JM Irish), NINDS 1R01 NS078289 (KC Ess), DOD W81XWH-16-1-0171/TS150037 (RA Ihrie), a research grant from the Tuberous Sclerosis Alliance (RA Ihrie), the VICC ACS-IRG 15-169-56 Pilot Project Grant (RA Ihrie), a grant from the VICC Michael David Greene Brain Cancer Research Fund, and a VICC Young Ambassadors Discovery Grant (RA Ihrie).

## Author Contributions

GV Rushing: conceptualization, funding acquisition, validation, investigation, visualization, methodology, and writing—original draft, review, and editing.
AA Brockman: investigation.
MK Bollig: investigation.
N Leelatian: visualization and methodology.
BC Mobley: resources.
JM Irish: supervision, visualization, and methodology.
KC Ess: resources, supervision, funding acquisition, and writing—review and editing.
C Fu: resources and funding acquisition.
RA Ihrie: conceptualization, supervision, funding acquisition, visualization, methodology, and writing—original draft, review, and editing.

## Conflict of Interest Statement

JM Irish is a co-founder and board member at Cytobank Inc. and received research support from Incyte Corp., Janssen, and Pharmacyclics. The other authors declare no conflict of interest.

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
