## [Reviewer comments · Life Science Alliance]

Life Science Alliance

Location-dependent maintenance of an intrinsic susceptibility to mTORC1-driven tumorigenesis

Gabrielle Rushing, Asa Brockman, Madelyn Bollig, Nalin Leelatian, Bret Mobley, Jonathan Irish, Kevin Ess, Cary Fu, and Rebecca Ihrie

DOI: <https://doi.org/10.26508/lsa.201800218>

Corresponding author(s): Rebecca Ihrie, Vanderbilt University

Review Timeline:	Submission Date:	2018-10-19
	Editorial Decision:	2018-10-22
	Revision Received:	2019-02-25
	Editorial Decision:	2019-02-26
	Revision Received:	2019-03-06
	Accepted:	2019-03-07

Scientific Editor: Andrea Leibfried

Transaction Report:

Please note that the manuscript was previously reviewed at another journal and the reports were taken into account in the decision-making process at Life Science Alliance. Since the original reviews are not subject to Life Science Alliance's transparent review process policy, the reports and author response cannot be published.

October 22, 2018

Re: Life Science Alliance manuscript #LSA-2018-00218-T

Dr. Rebecca Ihrie
Vanderbilt University Medical Center
Nashville, TN 37232

Dear Dr. Ihrie,

Thank you for transferring your manuscript entitled "Location-dependent maintenance of an intrinsic susceptibility to mTORC1-driven tumorigenesis" to Life Science Alliance. The manuscript was assessed by expert reviewers at another journal before, and the editors transferred those reports to us with your permission.

The reviewers pointed out that your conclusion on altered mTORC signaling strength in different NSPC populations would need further support with additional assays. They further noted that the TSC2 knock-out experiment requires a control, and that the effects observed in vivo could be confounded by the epilepsy mice develop. The reviewers also suggested to support the observed effects by performing in vitro proliferation assays. We think that these concerns can be addressed in a point-by-point response and text changes and by revising your work, and we would be happy to publish such a revised version in Life Science Alliance. We would thus like to invite you to revise your work, including an additional assay for mTORC signaling strength and the requested control, and - if feasible - an in vitro proliferation assay upon TSC2 knock-down. Please note that the request for neurosphere assays is not required for publication here.

Thank you for this interesting contribution to Life Science Alliance. We are looking forward to receiving your revised manuscript.

Sincerely,

- A letter addressing the reviewers' comments point by point.
- An editable version of the final text (.DOC or .DOCX) is needed for copyediting (no PDFs).
- High-resolution figure, supplementary figure and video files uploaded as individual files: See our detailed guidelines for preparing your production-ready images, <http://life-science-alliance.org/authorguide>
- Summary blurb (enter in submission system): A short text summarizing in a single sentence the study (max. 200 characters including spaces). This text is used in conjunction with the titles of papers, hence should be informative and complementary to the title and running title. It should describe the context and significance of the findings for a general readership; it should be written in the present tense and refer to the work in the third person. Author names should not be mentioned.

B. MANUSCRIPT ORGANIZATION AND FORMATTING:

Full guidelines are available on our Instructions for Authors page, <http://life-science-alliance.org/authorguide>

February 26, 2019

RE: Life Science Alliance Manuscript #LSA-2018-00218-TR

Dr. Rebecca Ihrie
Vanderbilt University
Cell and Developmental Biology
2220 Pierce Ave
761 PRB
Nashville, TN 37232-6840

Dear Dr. Ihrie,

Thank you for submitting your revised manuscript entitled "Location-dependent maintenance of an intrinsic susceptibility to mTORC1-driven tumorigenesis". We appreciate your point-by-point response and the introduced changes and we would thus be happy to publish your paper in Life Science Alliance pending final revisions necessary to meet our formatting guidelines:

- please provide the manuscript file as a word docx file
- please upload individual files for each figure (also for S figures)
- please include a statement that informed consent was obtained from all human subjects/their legal representatives and that the experiments conformed to the principles set out in the WMA Declaration of Helsinki and the Department of Health and Human Services Belmont Report
- please add callouts in the manuscript text for figure 4H, S2A, Table S1
- please note that some figure callouts are missing 'Fig' or 'Figure' (eg, "(S3A)" on page 5)
- please note that on p32, "(see antibody table)" needs to be more specific: "table S1"
- please note that Figure panel 4H is currently mis-spelled as '4G' in the figure
- I would like to suggest to include the methods and references of the supplemental information in the main manuscript

A. FINAL FILES:

B. MANUSCRIPT ORGANIZATION AND FORMATTING:

Sincerely,

Andrea Leibfried, PhD
Executive Editor
Life Science Alliance
Meyerhofstr. 1
69117 Heidelberg, Germany
t +49 6221 8891 502

e.a.leibfried@life-science-alliance.org
www.life-science-alliance.org

March 7, 2019

RE: Life Science Alliance Manuscript #LSA-2018-00218-TRR

Dr. Rebecca A Ihrie
Vanderbilt University
Cell and Developmental Biology
2220 Pierce Ave
761 PRB
Nashville, TN 37232-6840

Dear Dr. Ihrie,

Thank you for submitting your Research Article entitled "Location-dependent maintenance of an intrinsic susceptibility to mTORC1-driven tumorigenesis". It is a pleasure to let you know that your manuscript is now accepted for publication in Life Science Alliance. Congratulations on this interesting work.

DISTRIBUTION OF MATERIALS:

Again, congratulations on a very nice paper. I hope you found the review process to be constructive and are pleased with how the manuscript was handled editorially. We look forward to future exciting submissions from your lab.

Sincerely,

Andrea Leibfried, PhD
Executive Editor
Life Science Alliance
Meyerohofstr. 1
69117 Heidelberg, Germany
t +49 6221 8891 502
e a.leibfried@life-science-alliance.org
www.life-science-alliance.org